# Structural Optimisation and Design of a Cable-Driven Hyper-Redundant Manipulator for Confined Semi-Structured Environments

**DOI:** 10.3390/s22228632

**Published:** 2022-11-09

**Authors:** Rami Al-Khulaidi, Rini Akmeliawati, Steven Grainger, Tien-Fu Lu

**Affiliations:** Robotics and Automation Research Group, School of Mechanical Engineering, The University of Adelaide, Adelaide, SA 5005, Australia

**Keywords:** robotics in agriculture, hyper-redundant, redundant manipulators, confined environments, semi-structured environments, structural optimisation

## Abstract

Structural optimisation of robotic manipulators is critical for any manipulator used in confined semi-structured environments, such as in agriculture. Many robotic manipulators utilised in semi-structured environments retain the same characteristics and dimensions as those used in fully-structured industrial environments, which have been proven to experience low dexterity and singularity issues in challenging environments due to their structural limitations. When implemented in environments other than fully-structured industrial environments, conventional manipulators are liable to singularity, joint limits and workspace obstacles. This makes them inapplicable in confined semi-structured environments, as they lack the flexibility to operate dexterously in such challenging environments. In this paper, structural optimisation of a hyper-redundant cable-driven manipulator is proposed to improve its performance in semi-structured and challenging confined spaces, such as in agricultural settings. The optimisation of the manipulator design is performed in terms of its manipulability and kinematics. The lengths of the links and the joint angles are optimised to minimise any error between the actual and desired position/orientation of the end-effector in a confined semi-structured task space, as well as to provide optimal flexibility for the manipulators to generate different joint configurations for obstacle avoidance in confined environments. The results of the optimisation suggest that the use of a redundant manipulator with rigid short links can result in performance with higher dexterity in confined, semi-structured environments, such as agricultural greenhouses.

## 1. Introduction

The introduction of robotics in the middle of the twentieth century promoted the use of robots in many industrial and commercial automated contexts [1,2,3,4,5]. However, they are mostly used in fully-structured industrial environments [6,7], where the robot workspaces are clearly defined and known a priori. Only recently have robots been further utilised in semi-structured environments [8,9,10,11], such as in agriculture, in which the surroundings are dynamic and can be uncertain. Some work has presented strategies for introducing robotic manipulators into such environments [8,10,12,13]. Nevertheless, all the manipulators used are similar to conventional manipulators, with the 5-6 degrees of freedom (DOFs) that are found in a fully-structured industrial environment. Such robots have been proven to be inefficient, as the manipulators suffer from a lack of dexterity appropriate for the given workspace [10,14,15].

A semi-structured environment is defined as an environment with known landmarks that has no static topology, where the surroundings cannot be classified into, for example, rooms and walls [16]. It is instead a dynamic environment and subject to change during the operation of the manipulators [17]. This means any interactions with the environment require continuous replanning of the path for the end-effector to reach a certain point [18]. As robotic manipulators interact with dynamic environments, they should be sufficiently flexible to achieve any given task in their workspace. Conversely, a structured environment is an environment in which the manipulators repeat the same task in predefined and fully controlled regions [19,20]. An example of a conventional robotic manipulator that has been tested in a semi-structured environment is a bespoke platform developed to harvest lettuce [14]. The platform consists of a vision system with a convolutional neural network and custom end-effector software. The vision system shows a reasonable performance. However, the performance of the 6-DOF conventional robotic manipulator has proven to be inefficient due to its structural limitations, specifically the length of its links. As a result, although 69 lettuces could be detected by the vision system, only 31 lettuces were successfully harvested from the 60 that were within the range of the manipulator. This is due to the length of the links that connect every two consecutive joints, which proved too long to have the required dexterity to operate in the given workspace. This shows that conventional robotic manipulators are unable to provide the high degree of flexibility needed to work in semi-structured environments, which further raises the need for highly-dexterous manipulators that can move freely and flexibly within their workspace. This flexibility can be achieved using redundant/hyper-redundant manipulators, which have more degrees of freedom than are required to perform specific tasks in the workspace [21,22]. The inverse kinematics of such manipulators have an infinite number of solutions. This can be beneficial in confined semi-structured/unstructured environments, as the redundancy can be used for an infinite number of configurations without affecting the end-effector location [23].

The movement of manipulators in a confined semi-structured environment is determined by the positioning of the end-effector, which is the primary task, and the manipulator’s configuration, which is the movement of all joints, as a secondary task. Thus, in addition to the primary task (end-effector positioning), the redundancy of redundant/hyper-redundant manipulators is used to perform a secondary task, which changes the manipulator’s configuration in confined and enclosed spaces, such as in agriculture, construction, tooling, inspection, space exploration and underwater applications [24,25,26].

Robotic manipulators in confined semi-structured environments are typically required to perform tasks that are relatively complex, such as picking crops or fruit [27]. These environments constrain manipulators from moving freely due to dimensional limitations and the presence of obstacles, such as tree branches. To overcome the challenges and enable robotic manipulators to move freely and flexibly without damaging or hitting tree branches, fruit, or any other obstacles, structural optimisation should be conducted.

Some recent work on the structural optimisation of robotic manipulators has been presented to optimise the structure of serial and parallel manipulators by finding optimal link lengths [28,29]. For example, Bjorlykhaug et al. optimised the structure of a 6-DOF serial manipulator using Genetic Algorithms (GA) [30]. A generic method was proposed to optimise the structure of a redundant serial manipulator [31]; however, kinematics optimisation was only performed to evaluate the positioning of the end-effector, without considering other important measures, such as dexterity and manipulability.

Manipulability is critical for high dexterity [32]. Dexterity can be defined as the ability of the manipulator to reach a specific location in the workspace [33]. While the manipulability of a robot is its capacity to change the end effector’s position as a function of the joint configuration [34]. Manipulability is measured using a manipulability index [35,36,37], and the greater the manipulability index, the more dexterous the robot. The manipulability index value is affected by the number of joints and the lengths of the links. As dexterity has diverse definitions, in this article we define dexterity in semi-structured environments as the ability of the manipulator to reach a specific point using different configurations. Therefore, in the case of semi-structured environments, dexterity is highly dependent on manipulability because manipulability reveals if the manipulator can perform different configurations flexibly to reach a specific point.

Before implementing the robotic manipulators in semi-structured environments, their dexterity and manipulability must be considered in the optimisation process to ensure appropriacy and applicability. In the earlier work on semi-structured environments, the manipulability criteria, despite their critical role, have not been considered. The research to date only considered conventional industrial manipulators, which are not applicable in confined semi-structured environments due to the long distance between joints. The distance between every two consecutive joints should be optimised for better manipulability in confined semi-structured environments. Therefore, this article aims at extending the aforementioned works by optimising the structure of hyper-redundant manipulators to enable them to work flexibly with high manipulability within their given workspace in confined semi-structured environments. The introduced optimisation methods can result in an optimal design of robotic manipulators as the manipulability measures are considered in the optimisation process. This makes the optimised manipulator highly applicable in such confined environments as the manipulability measures are critical in terms of assessing the performance of robotic manipulators in this type of environment.

In this study, the robotic arm’s joint angles and link lengths are optimised using three manipulability measures and inverse kinematics optimisation to ensure the manipulator can move flexibly within its workspace with different configurations. Two algorithms are considered for the optimisation: Artificial Bees Colony (ABC) [38] and Fmincon [39], which is a built-in optimisation algorithm in MATLAB. Then, the optimisation results are validated using a hyper-redundant manipulator, which was created based on the optimisation results. The performance of the developed manipulator is tested in terms of manipulability by generating a collision-free trajectory in a confined semi-structured environment and analysing its manipulability at certain points within the environment. The results show that the optimised manipulator can achieve high manipulability even though the environment used for the validation is different from those used for optimisation. Based on the optimisation results, this implies that the manipulator can be used in any given confined semi-structured environment as it has high manipulability measures.

The remaining sections of this article are organised as follows: Section 2 presents the structural optimisation of a hyper-redundant manipulator to find the optimal length of links and joint angles that enable the manipulator to move with high manipulability. Section 3 shows the results of the proposed optimisation methods, along with the validation of the results. The conclusion is drawn in Section 4.

## 2. Materials and Methods

### 2.1. Structural Optimisation of Hyper-Redundant Manipulators

Optimisation is the process of determining the optimal solution among diverse alternatives; it is necessary for robotic manipulator design so that the optimal structure of joint parameters can be obtained. Several optimisation algorithms are available, including Genetic Algorithms (GA), Particle Swarm Optimisation (PSO) and Artificial Bees Colony (ABC). No optimisation algorithm can be classified as the unqualified best, as the performance of each specific algorithm depends on the problem structure, which means some objective functions can perform faster than others, depending on the problem. However, ABC generally showed successful and stable results when compared with GA and PSO when it was tested to minimise the makespan for job scheduling, and protein production problems [40,41], as it showed advantages in stability and not being trapped in local minima. Moreover, it does not employ external parameters such as crossover and mutation rates, as in GA [42]. Another widely used optimisation solver is the Fmincon solver; it is the main optimisation solver in the MATLAB Optimisation Toolbox [39], which performs better for nonlinearly constrained problems. Fmincon is based on interior point techniques for nonlinear minimisation. The flowcharts of both optimisation algorithms are shown in Figure 1 and Figure 2.

Any optimisation problem can be defined as finding the vector of parameters that minimises or maximises an objective function:(1)$$\min\;\mathrm{or}\;\max f\left(z\right)=\left(z_{1},\;z_{2},\ldots ,\;z_{n}\right)\;$$
and constrained by:$$z_{min}\leq z_{i}\leq z_{max},\;i=1,\;2,\ldots ,\;n$$
where z represents the vector consisting of parameters to be optimized. In this study, $z_{i}$ is defined as $x_{i}={\left[q,a\right]}^{T}$, where $q=\left(q_{1},\;q_{2},\ldots ,\;q_{n}\right)$ is the joint angles in degrees, and a is the link length of a hyper-redundant manipulator in mm. In this case, all the links are assumed to have an equal length. The objective function used in this study to optimise the joint parameters for *n* DOF is given by:(2)$$\left[\;\begin{array}{c} \begin{array}{c} \;max\;f\left(x\right),\;\\ x=\left[q,a\right], \end{array}\\ \;s.t.\;\underline{q}\leq q_{i}\leq \bar{q}\\ \;\underline{a}\leq a\leq \bar{a}\\ \;i=1,2,\ldots ,n \end{array}\right]$$
where f(.) is the manipulability measure, which is used as the objective function, and $\underline{q}$, $\bar{q}$ and $\underline{a}$, $\bar{a}$ are the lower bound and upper bound for joint angles and the length of the link, respectively. When choosing lower and upper bound values, the joint structure and limitations were considered, as well as the designs in previous studies that showed a lack of flexibility to work in their given workspace [14,24,43]. Thus, the lower bound was chosen to avoid being too small to provide sufficient space for other components, such as pulleys, wires and joint sensors, and the upper bound was sufficiently large that its size would not affect its manipulability. That is, if the optimized length of the link were too small, pulleys, wires and sensors cannot be added to the joint due to space limitations; while, if it is too long, it will result in a manipulator that is not flexible enough to handle its workspace as with those in the aforementioned studies.

For the proposed redundant manipulator, in order to achieve high manipulability and dexterity, the optimised objective functions are based on manipulability measures and inverse kinematics. Based on the definitions of dexterity and manipulability, the objective functions are dependent on the joint parameters. The minimum and maximum values used in both optimisation algorithms are ±90 degrees for joint angles and 80 mm and 250 mm for the length of the link, respectively. The minimum value is chosen, as mentioned above, to provide sufficient space for the joint components, whilst the dexterity performance is used to define the maximum value as the link length, for which greater than 250 mm was proven to lack dexterity in any workspace [14].

The manipulator includes universal joints, as described further in Section 3. Thus, every joint is driven by two motors–one for each axis. The joint is attached to two pulleys that are connected to the motors by cables to enable each motor to drive one axis. The last joint is considered a general end-effector so that it can be incorporated into any type of bespoke gripper. Another aspect of the design is the base, which houses the motors, electronics and power equipment. The cables that are used in the design are of the Bowden cable type and are assumed in this work to have no effect on the kinematics and dynamics.

### 2.2. Optimisation Algorithms

The first optimisation algorithm used in our work to determine the optimal joint parameters is ABC as it can avoid being trapped in local minima wherein different joint parameters are tested and compared. ABC is a metaheuristic method inspired by foraging honeybees. There are three groups of bees in ABC: employed bees, onlooker bees and scout bees. Onlooker bees and scout bees are also called unemployed bees. Initially, the scout bees look for food until they find its source. Then, the employed bees and onlooker bees exploit the abundance (fitness) of the food source. The employed bees are linked to a certain food source and share information about the food by means of dancing. Their dancing is observed by onlooker bees to determine the best source using probability. The location of the food source reflects a solution in the ABC algorithm, and the abundance of food represents the fitness of the solution.

For the proposed structural optimisation, the initialisation stage of the algorithm starts by assigning initial values to the joint angles and length of the link. Then, it initialises the population and the limits of the variables. Next, it starts by assigning the initial solution, which is, in this study, zero for manipulability optimisation, with infinity as the distance between the current and desired end-effector coordinates. The corresponding position for every employed bee represents the values of the joint parameters. Abundance is shown by the manipulability index and the distance between the current and desired position of the end-effector.

After the initialisation process, the employed bees start searching for a new food source $v_{m}$, which is more abundant than $x_{m}$, which is already in their memory, and evaluate its *i*-th fitness using the following equation [38]:(3)$$v_{mi}=x_{mi}+\varphi_{mi}\left(x_{mi}-x_{ki}\right)$$
where i is a randomly selected parameter index, $x_{k}$ is a randomly chosen food source, and $\varphi_{mi}$ is chosen randomly between −1 and 1. Then, after the new food source is identified using Equation (3), the fitness of the new food source is calculated, and a greedy selection is performed between $v_{mi}$ and $x_{mi},$ as follows:(4)$$fit_{m}\left(x_{m}\right)=\{\begin{array}{cc} \frac{1}{1+f_{m}\left(x_{m}\right)}, & \;f_{m}\left(x_{m}\right)\geq 0\\ 1+abs\left(f_{m}\left(x_{m}\right)\right), & \;f_{m}\left(x_{m}\right)<0 \end{array}$$
where $fit_{m}\left(x_{m}\right)$ is the fitness of the new food source, and $f_{m}\left(x_{m}\right)$ is the objective function value for the solution $x_{m}$. Then, the information is shared by the employed bees with the onlooker bees, which determine the best solution among the fitness values using probability, $p_{m}$. The best solution is determined using the roulette wheel fitness selection method [44], as in Equation (5):(5)$$p_{m}=\frac{f_{m}\left(x_{m}\right)}{\sum_{m=1}^{SP}f_{m}\left(x_{m}\right)}$$
where SP is the size of the population, which was chosen, based on several trials, to be 500, divided equally into employed and onlooker bees so the employed bees share the information about their food source to recruit the corresponding onlooker bees. Based on the shared information, the onlooker bees choose the food source. Then, the employed bees, whose solution cannot be improved after certain trials, abandon their solution using abandonment criteria. As a result, they become scout bees, looking for a new source of food. The abandonment criteria define the control limit for jumping from local minima to global minima; that is, if the solution of the employed bees does not change after a number of iterations, they are converted to scout bees. The overall flowchart of the algorithm is shown in Figure 1.

The number of iterations in ABC is 300 for manipulability measures and 1000 for inverse kinematics. The objective function values for manipulability measures did not increase from around 250 iterations, which means the best value is reached at around 250 iterations. This was observed after several trials when testing the algorithm. Therefore, the number of iterations was reduced to 300 to save computational time. In Fmincon, on the other hand, the initial number of iterations was set to 5000, and the exit criteria depend on the tolerance, which is 1e-16. This means that although the number of iterations was set too high, the optimisation can stop after a few iterations as long as the objective function value is not increasing, as the results show in the next section.

If, for example, the algorithm starts by assigning the number of joint angles, based on the given length of the manipulator workspace, the length of the links is identified. The next step is identifying the distance between every two consecutive joints at each iteration. Then, the algorithm starts by assigning the minimum number of joints and checking the manipulability and dexterity with iteratively assigned joint parameter values and storing the value of the corresponding objective function. Next, it assigns a different number of joint parameters and evaluates their objective function and compares the value of the current objective function with the previous objective function value. As the manipulability depends on the value of the specific joint angles, the subroutine has to iteratively assign different angles and check the manipulability measures.

Finally, after considering the optimisation results, a computer-aided design (CAD) model of the manipulator will be created. Every link should have two axes (pitch and yaw) so the manipulator can move freely and flexibly and cover the area within the range of the manipulator. As highlighted previously, the main reason for the low level of dexterity is the long length of the links. In other words, every consecutive similar motion (for example, yaw-yaw) should be close to one another, based on the optimised values of each joint’s parameters, so the manipulator will be sufficiently dexterous for the specific task.

### 2.3. Kinematics of Hyper Redundant Manipulators

The optimisation in this study is based on inverse kinematics and manipulability. The different manipulability measures used for optimisation are based on the Jacobian matrix. The kinematics of the proposed hyper-redundant manipulator are solved using Denavit-Hartenberg (DH) parameters, as shown in Table 1. The model consists of n revolute joints along the *z*-axis of frames {1} to {n}. As the manipulator is a universal joint-type manipulator, n is an even number. The forward kinematics function, *h*(.), is:(6)y=h(q) 
where $y={\left(x,y,z,\;\theta_{x},\;\;\theta_{y},\;\;\theta_{z}\right)}^{T}$ is the pose vector (the translational pose in the *x*, *y* and *z* axes, and the rotational angles with respect to the *x*, *y* and *z* axes) in the task space, and $q={\left(q_{1},\;\;q_{2},\ldots ,\;q_{n-1},\;q_{n}\right)}^{T}$ is the joint angle vector in the configuration space.

The transformation between the base frame and any other frame is given by the transformation matrix as:(7)$$T=\left(\begin{array}{cccc} \cos\left(q_{n}\right) & -\cos\left(\alpha_{n}\right)\;\sin\left(q_{n}\right) & \sin\left(\alpha_{n}\right)\;\sin\left(q_{n}\right) & a_{n}\;\cos\left(q_{n}\right)\\ \sin\left(q_{n}\right) & \cos\left(\alpha_{n}\right)\;\cos\left(q_{n}\right) & -\sin\left(\alpha_{n}\right)\;\cos\left(q_{n}\right) & a_{n}\;\sin\left(q_{n}\right)\\ 0 & \sin\left(\alpha_{n}\right) & \cos\left(\alpha_{n}\right) & d_{n}\\ 0 & 0 & 0 & 1 \end{array}\right)$$
where $d_{n}$ is the offset along the $Z_{n-1}$ axis; a***_n_*** is the length of the common normal, which in our case is the length of the link connecting the previous yaw movement axis to the next pitch movement axis; and $\alpha_{n}$ is the angle about the common normal, from the $Z_{n-1}$ axis to the Z***_n_*** axis.

The inverse kinematics are used to find the function $h^{-1}$, which maps the task space to the configuration space:(8)$$q=h^{-1}\left(y\right)\;$$

The configuration space of this type of manipulator is larger than the task space, which results in infinite inverse kinematics solutions with v number of equations and u unknowns, where v>u, with u=6, which corresponds to 3 translation and 3 angular motions in a 3D space. Therefore, iterative methods using the Jacobian matrix are used, as analytical solutions do not exist. Deriving Equation (6) with respect to time gives:(9)$$\dot{y}=J\dot{q}$$
where J is the Jacobian matrix that maps the translational and the angular velocities of the joints to the pose velocities.
(10)$$J=\left(\begin{array}{ccc} \frac{\partial y_{1}}{\partial q_{1}} & \cdots & \frac{\partial y_{1}}{\partial q_{v}}\\ \vdots & \ddots & \vdots \\ \frac{\partial y_{u}}{\partial q_{1}} & \cdots & \frac{\partial y_{u}}{\partial q_{v}} \end{array}\right)$$

As the inverse of the Jacobian matrix can only exist for non-singular square matrices, the Moore-Penrose pseudoinverse is taken to give:(11)$$\dot{q}=J^{+}\dot{y}$$
where $J^{+}$ is the pseudoinverse of the Jacobian matrix, which is equal to:(12)$$J^{+}=J^{T}{\left(J\;J^{T}\right)}^{-1}$$
where $J^{T}$ is the transpose of the Jacobian matrix.

## 3. Results and Discussion

### 3.1. Manipulability Optimisation

Manipulability optimisation has been undertaken to ensure a high manipulability measure. Three different manipulability measures were used as objective functions to optimise the best joint parameter values. The first measure used is the Yoshikawa manipulability measure [32], which is given by:(13)$$µ=\sqrt{det\left(J\;J^{T}\right)}$$
where $\det\left(J\;J^{T}\right)$ is the determinant of the multiplication of the Jacobian matrix and its transpose. The Yoshikawa measure is equal to the product of the singular values of the Jacobian matrix. As this index is proportional to the velocity ellipsoid, the high singular values mean a high end-effector velocity for the unit norm: a manipulator is considered to have a high manipulability if its velocity ellipsoid is isotropic, i.e., has a sphere-shape, which means it can change its end-effector position in all three axes equally. The objective function values and the optimised joint parameters for the optimisation based on the Yoshikawa manipulability measure using Fmincon and ABC are shown in Figure 3 and Figure 4.

However, the Yoshikawa measure is not an accurate manipulability measure, as the velocity ellipsoid cannot depict the exact status of every singular value. For example, if one singular value is very small, the velocity ellipsoid can still be large as the other singular values are high. Furthermore, this measure is unbounded: therefore, another measure was used to compensate for the downside of the Yoshikawa manipulability measure: the Local Conditioning Index (LCI). The LCI is the ratio of the minimum and maximum singular values [45,46], Equation (14):(14)$$\mathrm{LCI}=\frac{\sigma_{min}}{\sigma_{max}},\;\in \left[0,1\right]$$
where $\sigma_{min}$ and $\sigma_{max}$ are the minimum and maximum singular values, respectively.

The result obtained from the LCI manipulability measure as an objective function for parameter optimisation is shown in Figure 5 and Figure 6:

The third manipulability measure that was used to optimise the parameters is a new measure introduced by Mariass et al. [47]. This measure is bounded and can be written as a function of the joint coordinates. The measure is given by:(15)$$\Delta H=\frac{H}{A},\;\in \left[0,1\right]$$
(16)$$H=\frac{u}{tr\left({\left(J\;J^{T}\right)}^{-1}\right)}\;$$
(17)$$A=\frac{\;tr\left(J\;J^{T}\right)}{u}\;$$
where H and A are the harmonic and arithmetic mean of eigenvalues, respectively, and tr is the trace of the matrix. Figure 7 and Figure 8 show the result for the ΔH measure when used as an objective function for parameter optimisation.

It can be seen from the results of the three manipulability measures that the optimisation algorithms continued to reduce the links’ length until it reached the lower bound limit of length between every two joints, resulting in a 16-DOF manipulator. This means the shorter the length of links, the better the manipulability that can be achieved. However, in practice, the length of the link should be sufficient to avoid design limitations by providing space for joint sensors, cables and pulleys. Hence, the lower bound of the link lengths is restricted to 80 mm.

### 3.2. Inverse Kinematics (IK) Optimisation

Inverse kinematics optimisation was conducted to verify the performance of the manipulator, considering certain task space coordinates. It is based on the results from the manipulability optimisation in Section 2.2 in terms of the optimal number of joints. The objective function and constraints for joint parameters in this part are similar to Equation (2), with G(.,.,.), Equation (18), as the objective function, which minimises the error between the current and desired positions of the end-effector. Additional workspace constraints were added to ensure the manipulator can perform flexibly in confined and enclosed workspaces. The workspace was constrained to resemble a box with dimensions of 1000 mm, 700 mm and 900 mm in the *x*, *y* and *z* coordinates, respectively. This means the end-effector should not go beyond this workspace. Another constraint was added to limit the middle joint to spaces no larger than 500 mm, 350 mm and 450 mm in the *x*, *y* and *z* coordinates, respectively.
(18)$$G\left(x,y,z\right)\;=\sqrt{{\left(x-x_{d}\right)}^{2}+{\left(y-y_{d}\right)}^{2}+{\left(z-z_{d}\right)}^{2}}$$
where x and $x_{d}$ are the current and desired end-effector positions in the *x*-axis, respectively. Similarly, y and $y_{d}$, and z and $z_{d}$ are the current and desired end-effector coordinates in the *y*-axis and *z*-axis, respectively. The desired coordinates were 500 mm, 300 mm and 200 mm in the *x*, *y* and *z* coordinates, respectively. The result obtained by Fmicon for inverse kinematics optimisation was more accurate than the result obtained by ABC, as the error was nearly 0.002 mm using the Fmincon solver, while it was about 0.4 mm in ABC, as shown in Figure 9 and Figure 10:

All the optimised values of parameters obtained by the different objective functions used in this study are summarised in Table 2.

Based on the results of the different objective functions, either from manipulability measures or inverse kinematics, it is observable that the best performance of the manipulator can be obtained within a range between 90 mm and 150 mm for the length of the links. The joint angles also reached the lower and upper bounds of ±90 degrees. Therefore, considering the obtained results, the manipulator designed in this work has a length of 100 mm between every two consecutive similar joints, which satisfies the high dexterity needs of the manipulator.

### 3.3. Results Validation

Based on the optimisation results, a model for the optimised manipulator was developed using Autodesk Inventor, as shown in Figure 11. It consists of the housing for the actuators and electronic equipment, along with the manipulator. To account for dexterity, the universal joint is considered in the design, as shown in Figure 11 (right). The joint consists of two identical yokes joined by an octagon in the centre. Every yoke is connected to a pulley so it can perform two-axis (pitch and yaw) movements. The distance between every two consecutive joints is 100 mm, and the distance from the last joint to the tip is also 100 mm, as shown in Figure 12; the inner diameter of the joint is 60 mm, while its outer diameter is 90 mm. The joints’ movements are animated, and they can move ±90 degrees in both the pitch and the yaw axes.

To evaluate the dexterity and manipulability of the proposed manipulator, a confined semi-structured environment was created, as shown in Figure 13. It is a different environment from the environment used in the optimisation process to show that the manipulator, based on the optimisation results, can perform effectively in any confined semi-structured environment. The dimensions of the environment are 1.2 m in the *x*-axis, 0.6 m in the *y*-axis and 0.55 m in the *z*-axis. The environment consists of various shapes of obstacles including two boxes, two spheres and a cylinder. The first box is located at 0.4 m, 0.2 m and 0.2 m, in the *x*, *y* and *z* axes, respectively, while the second box is placed at 0.9 in the *x*-axis, 0.2 in the y-axis and 0.1 in the *z*-axis. The first box has a length of 0.2 m, a width of 0.1 m and a height of 0.15 m, while the second box is 0.3 m, 0.3 m and 0.08 m in length, width and height, respectively. Two spheres with a radius of 0.08 m were placed in the environment. The first one was positioned at [0.6 −0.1 0.35] in the [*x*
*y*
*z*] axes, while the second one was at [0.6 0.35 0.5] in the [*x*
*y*
*z*] axes. The cylinder had a radius of 0.05 m and a length of 0.25 m; it was located at 0.75, 0.1 and 0.45 in the *x*, *y* and *z* axes, respectively.

Next, a collision-free trajectory was generated to drive the manipulator from the home configuration, Figure 13 (right), at an end-effector pose of 1.026 m, 0 m and 0 m at the *x*, *y* and *z* axes respectively, to a final pose of 0.803 m in the *x*-axis, 0.154 m in the *y*-axis and 0.398 m in the *z*-axis, which is a location that is slightly behind the cylinder. Four waypoints were generated between the home configuration and the final pose. The middle way point was behind the second sphere at coordinates of 0.66 m in the *x*-axis, 0.29 m in the *y*-axis and 0.47 m in the *z*-axis. The middle waypoint and final pose configurations of the manipulator are reflected in Figure 14. The joint position and velocity trajectories are shown in Figure 15. Considering the restrictions of the confined workspace, the joint positions did not go to their maximum limit. The highest joint position was for joint one with 52.4 degrees, followed by joints nine, seven and eight with 40.7, 35.5 and −34.4 degrees, respectively, while joints four and eleven did not go above −28 degrees. Seven joints (two, six, ten, thirteen, fourteen, fifteen and sixteen) did not move at all, while joints three, five and twelve remained at almost zero throughout the trajectory. Considering the various obstacles in the given environment, the manipulator was able to move around obstacles and reach its goal without using the ±90-degrees full range of angles. Thus, implementing the manipulators resulting from the optimisation used in this study in environments with a different set of obstacles has been shown to result in a flexible performance given the high manipulability results.

Figure 16 shows the velocity ellipsoids for the manipulator, with the design based on the optimisation results, at the middle and final pose configurations. The figure shows that the proposed manipulator has a high manipulability as the velocity ellipsoids are both isotropic (have a spherical shape). This implies that the manipulator can change its end-effector positions in all axes, using different configurations of joints, despite the restrictions imposed by the confined workspace. Therefore, the manipulator, with its design being based on the optimisation results, can move freely and generate high velocity in the *x*, *y* and *z* axes, regardless of the shape of the obstacles.

## 4. Conclusions

This paper proposed a concept for structural optimisation of hyper-redundant manipulators that can be used in confined semi-structured environments, such as in agriculture. Two methods of optimisation were used, using the Fmincon and Artificial Bees Colony (ABC) algorithms, considering the manipulability measures and inverse kinematics, to ensure optimal joint parameters that enable the manipulator to work flexibly within its task space.

This optimisation method was conducted for manipulators with universal joints, as the universal joints provide a wide range of motion for the manipulator to cover two axes with a 360-degree workspace, so all points within the manipulator workspace are reachable. The optimised manipulator was assumed to have a 1 m length, and the optimisation method was conducted to ensure high performance within its workspace, so the reachability of the points within its workspace with the optimised optimal lengths of the links was shown to be successful. Therefore, for long manipulators, an additional number of links alone need to be added to the manipulator for it to cover its workspace dexterously.

A hyper-redundant manipulator was designed based on the results of optimisation to be used for validation in a confined semi-structured environment. Its performance was tested by assessing its manipulability in certain confined locations. It shows a high level of dexterity using different joint configurations, as it is depicted by manipulability ellipsoids. Unlike previous studies that only optimised conventional manipulators, which considered kinematic constraints alone, the optimisation in this study depends on the manipulability measures and inverse kinematics in a confined workspace. This shows the manipulator is dexterous and performs with high manipulability within any given workspace, as is shown as the testing environment was completely different from that in the optimisation scenario. This is demonstrated by the results of the validation, as the manipulator was able to reach its goal and move around the different obstacles with a range of angles between ±60 degrees, which proves the high level of flexibility of the optimised manipulator in handling its workspace. Therefore, more complicated workspaces with different obstacles can be handled within the optimised manipulator as its range of angles is ±90 degrees. This means the manipulator can perform flexibly within any given confined semi-structured environment or workspace. To the best of our knowledge, this is the first work that optimises hyper-redundant manipulators in semi-structured environments. The previous works only consider conventional manipulators, and dexterity and manipulability measures have been neglected in their optimisation. As a future extension of this study, various control techniques will be investigated to ensure the accuracy and robustness of the manipulator’s performance in semi-structured environments.

## Figures and Tables

**Figure 1 sensors-22-08632-f001:**
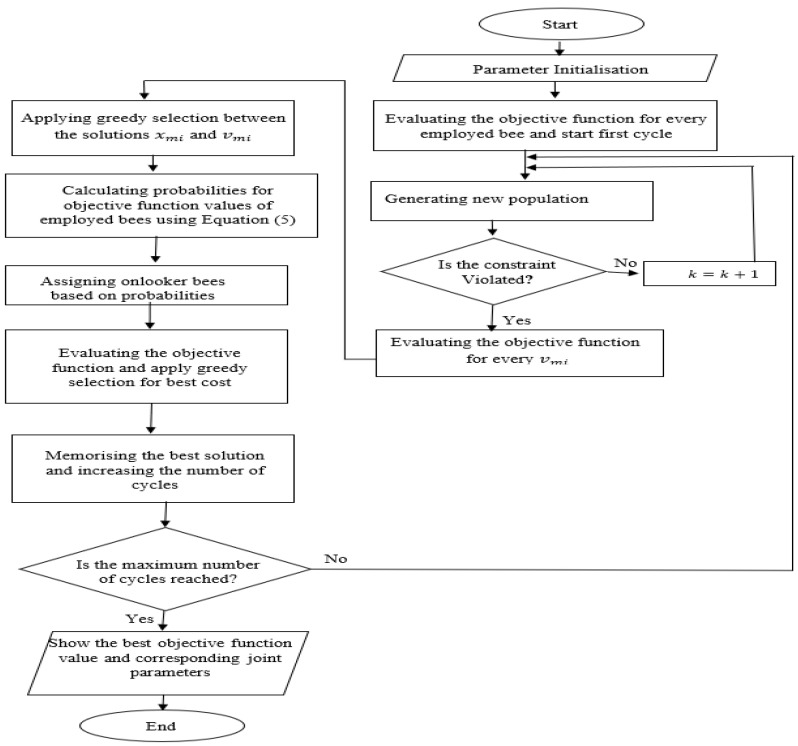
Flowchart of the Artificial Bees Colony algorithm.

**Figure 2 sensors-22-08632-f002:**
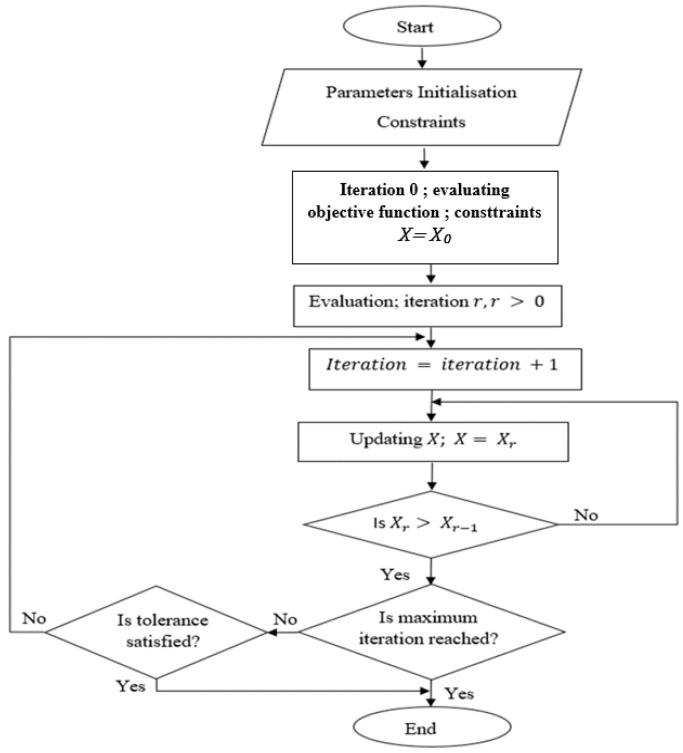
Flowchart of the Fmincon optimisation algorithm.

**Figure 3 sensors-22-08632-f003:**
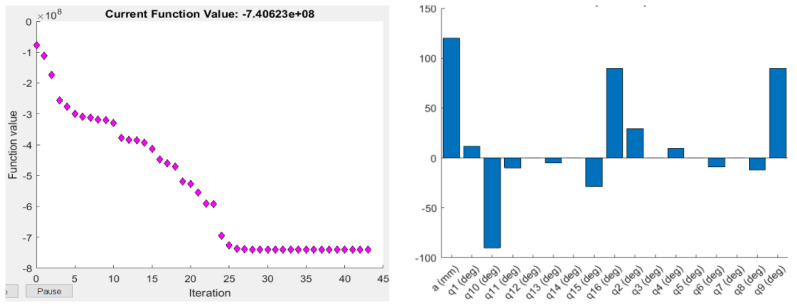
Objective function values (**left**), and the corresponding optimised joint parameters (**right**) for the best objective function value obtained by Fmincon for the Yoshikawa manipulability measure.

**Figure 4 sensors-22-08632-f004:**
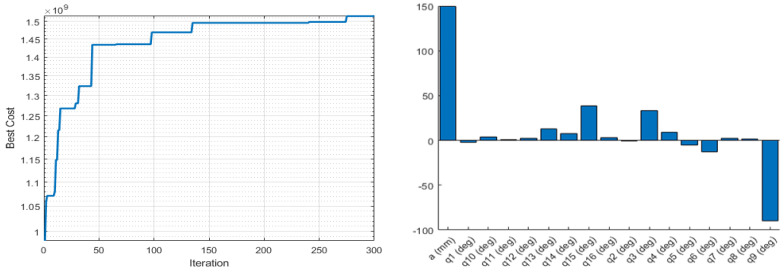
Objective function values (**left**), and the corresponding optimised joint parameters (**right**) for the best objective function value obtained by ABC for the Yoshikawa manipulability measure.

**Figure 5 sensors-22-08632-f005:**
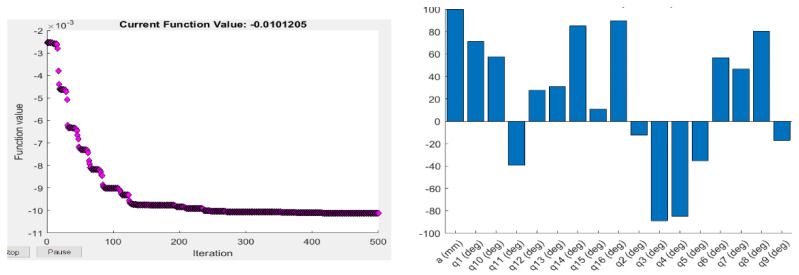
Objective function values (**left**), and the corresponding optimised joint parameters (**right**) for the best objective function value obtained by Fmincon for the LCI manipulability measure.

**Figure 6 sensors-22-08632-f006:**
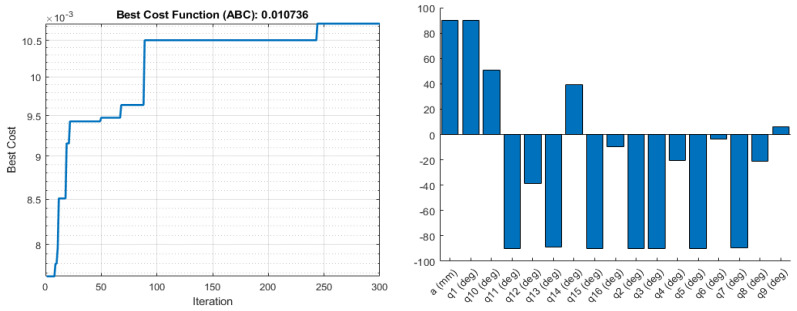
Objective function values (**left**), and the corresponding optimised joint parameters (**right**) for the best objective function value obtained by ABC for the LCI manipulability measure.

**Figure 7 sensors-22-08632-f007:**
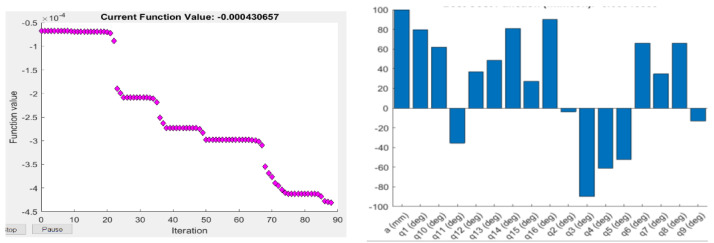
Objective function values (**left**), and the corresponding optimised joint parameters (**right**) for the best objective function value obtained by Fmincon for the ∆H manipulability measure.

**Figure 8 sensors-22-08632-f008:**
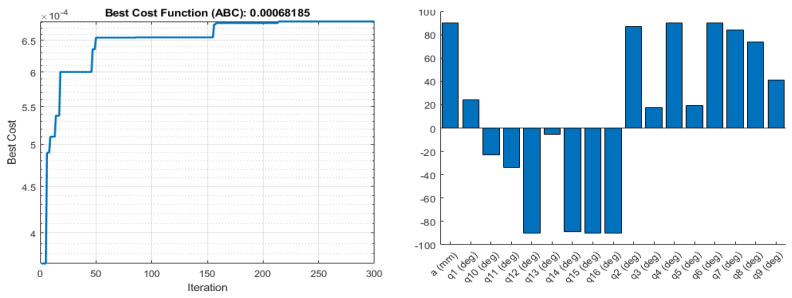
Objective function values (**left**), and the corresponding optimised joint parameters (**right**) for the best objective function value obtained by ABC for the ∆H manipulability measure.

**Figure 9 sensors-22-08632-f009:**
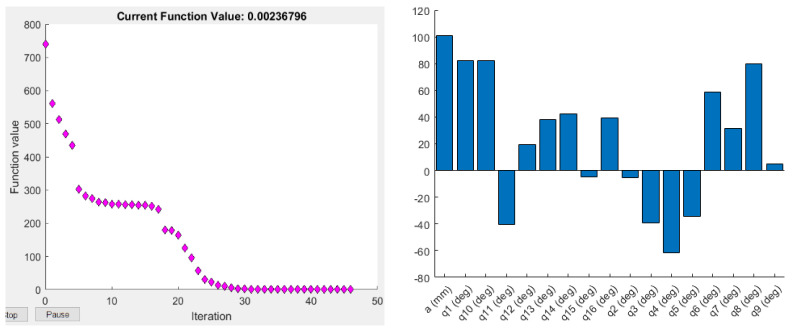
Objective function values (**left**), and the corresponding optimised joint parameters (**right**) for the best objective function value obtained by Fmincon for the IK optimisations.

**Figure 10 sensors-22-08632-f010:**
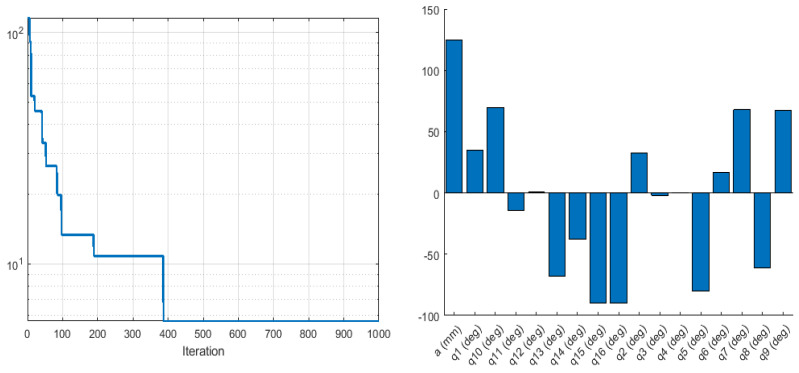
Objective function values (**left**), and the corresponding optimised joint parameters (**right**) for the best objective function value obtained by ABC for the IK optimisations.

**Figure 11 sensors-22-08632-f011:**
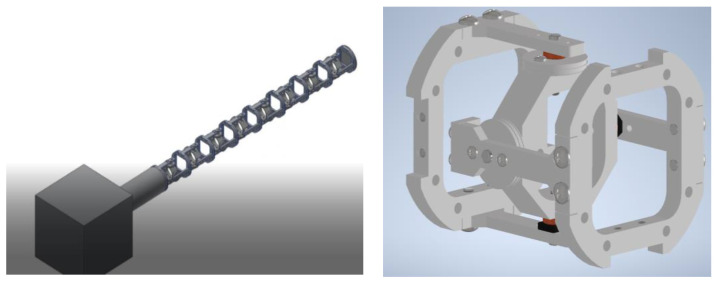
The overall structure of the manipulator (**left**) and the design of the joint (**right**).

**Figure 12 sensors-22-08632-f012:**
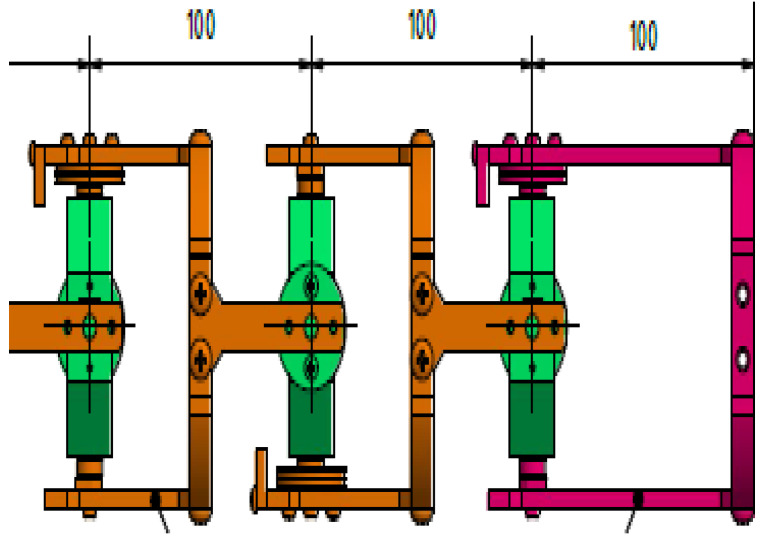
The last three joints’ structure.

**Figure 13 sensors-22-08632-f013:**
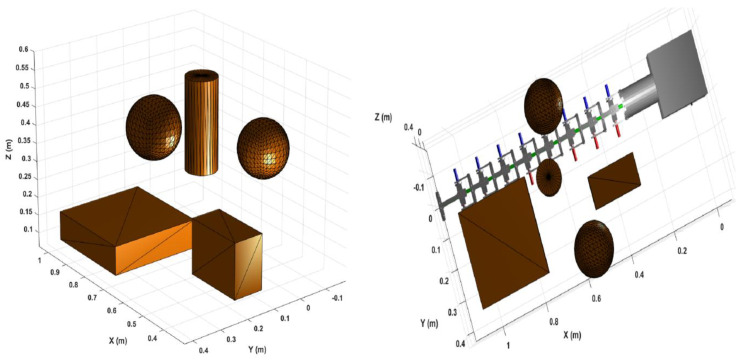
Enclosed semi-structured environments with the obstacles’ coordinates (**left**) and the manipulator’s home configuration (**right**).

**Figure 14 sensors-22-08632-f014:**
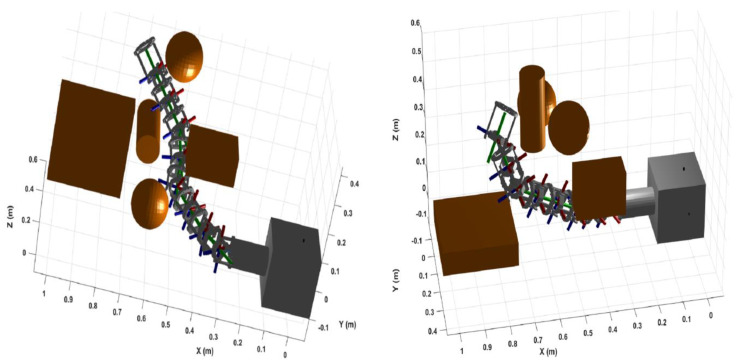
The manipulator configuration at the middle waypoint (**left**) and the final pose (**right**).

**Figure 15 sensors-22-08632-f015:**
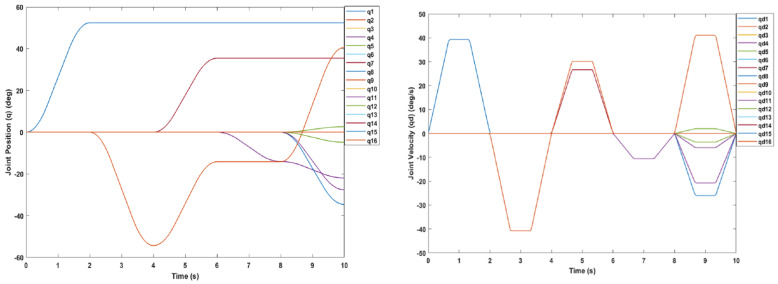
Joint angle trajectory positions (**left**) and velocity (**right**) from the home configuration to the final pose.

**Figure 16 sensors-22-08632-f016:**
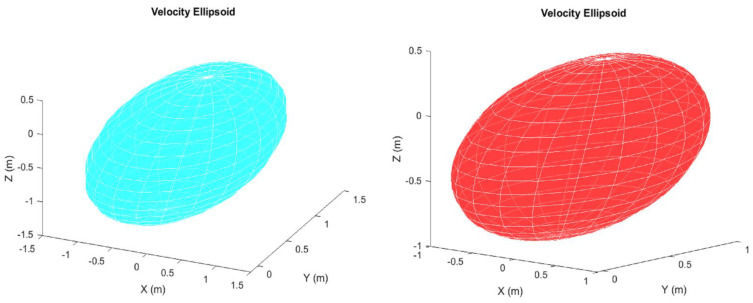
Velocity ellipsoid of the joint angles corresponding to the middle waypoint configuration (**left**) and final pose configuration (**right**).

**Table 1 sensors-22-08632-t001:** DH parameters.

Joint	$q_{n}$	$d_{n}$	$a_{n\;(\mathrm{mm})}$	$\alpha_{n\;(\deg)}$
1	$q_{1}$	$d_{1}$	$a_{1}$	$\alpha_{1}$
2	$q_{2}$	$d_{2}$	$a_{2}$	$\alpha_{2}$
. . .	. . .	. . .	. . .	. . .
n−1	$q_{n-1}$	$d_{n-1}$	$a_{n-1}$	$\alpha_{n-1}$
n	$q_{n}$	$d_{n}$	$a_{n}$	$\alpha_{n}$

**Table 2 sensors-22-08632-t002:** The optimal joint parameters corresponding to the best objective functions.

Joint Parameters	Manipulability Measures						Inverse Kinematics Optimisation	
	Yoshikawa		LCI		∆H			
	Fmincon	ABC	Fmincon	ABC	Fmincon	ABC	Fmincon	ABC
**Objective Function Value**	**7.4 × 10^8^**	**1.5** **× 10^9^**	**0.0101**	**0.0107**	**4.3 × 10^−4^**	**6.8 × 10^−4^**	**4.14 × 10^−6^**	**0.46**
**q_1_ (deg)**	11.4	−2.22	71	90.0	79.49	24.2	30.5	35
**q_2_ (deg)**	29.3	−1	−12	−90.0	−3.67	87.1	25.6	32
**q_3_ (deg)**	0	33.6	−88.7	−90.0	−89.91	17.8	14.6	−2
**q_4_ (deg)**	9.3	8.91	-84.9	-20.6	−61.2	90.0	−17	−0.3
**q_5_ (deg)**	0	−5.49	−35	−90.0	−52.3	19.5	−16	−80
**q_6_ (deg)**	−9.2	−12.6	56.8	−3.83	66	90.0	−19	16
**q_7_ (deg)**	0	2.02	46.6	−89.2	34.7	83.8	54	67
**q_8_ (deg)**	−12.2	1.89	80.3	−21.2	65.8	73.8	13.8	−61
**q_9_ (deg)**	90	−90.0	−17	6.3	−13	41.4	65.6	67
**q_10_ (deg)**	−90	3.62	57.5	50.7	62	−22.8	78	70
**q_11_ (deg)**	−9.8	0.487	−39.4	−90.0	−35.5	−33.6	70.5	−14.6
**q_12_ (deg)**	0	2.32	27.7	−38.6	37	−90.0	71.6	0.5
**q_13_ (deg)**	−4.7	13.3	30.8	−88.9	48.4	−5.19	−8.8	−68
**q_14_ (deg)**	0	7.36	85.18	39.5	81	−88.5	68	−38
**q_15_ (deg)**	−29	38.4	10.7	−90.0	27.3	−90.0	−37	−90
**q_16_ (deg)**	90	3.27	89.67	−9.72	90	−90.0	17	−90
**a (mm)**	120.00	150.0	100	90.0	100.0	90.1	107	124

## Data Availability

The data presented in this study are available on request from the corresponding author.

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
