# Peer review of "Structural Optimisation and Design of a Cable-Driven Hyper-Redundant Manipulator for Confined Semi-Structured Environments"

_sensors, 2022, doi:10.3390/s22228632_

Round 1
Reviewer 1 Report
The authors present an interesting paper about Strutural optimization applied to a hyper-redundant manipulator. however they shoud improve some (few) issues, namelly:
1. why references 21-22 & 30-35 are at the end of paper and they are not mentionated across the text? Are they important or not?
2. the conclusions have a very qualitative approach, so the authors should make a more numerical approach and even, as their approach/method has better results than other mehods.
3. As for the rest of the paper, it is well strutured and well written.
Finally, congratulation on your work.
Reviewer 2 Report
The authors have addressed the area of design optimization of manipulators for application in confined semi-structured environments especially in agriculture. The proposed method optimises the manipulator design to increase efficiency maximising dexterity for the given workspace. The method and results are well presented. However, following are comments to improve quality of the manuscript:
-
In abstract, the motivation of work is missing and needs to be clearly mentioned.
-
Few sentences are not clear. For example, line 158-162. Rephrase such sentences to make them understandable.
-
The English language is confusing at a few places in the manuscript. For example, line 170 (to defined?), future tense is used in section 2.2 etc.
- What is fitm in equation (3)?
Reviewer 3 Report
The paper proposes structural optimization of ultra-redundant cable-driven manipulators to improve their performance in semi-structured and challenging confined spaces. This need has some relevance in agricultural scenarios.
1. First of all the narrative structure of this paper is problematic and the research object is not given until the end, which is obviously unreasonable. This narrative would be fine if the authors proposed a new optimization algorithm and just used the manipulator structure as an optimization case, but obviously the focus of the paper is clearly not on the optimization algorithm, but on the manipulator structure optimization itself.
2. From the optimization method, there is nothing innovative, and since this is the case the paper should provide an in-depth discussion of structural optimization, such as optimization goals or constraints for different environmental parameter configurations, or manipulator dynamics or kinematic flexibility, etc. Therefore, the paper looks more like a report than an academic paper.
It is recommended that the authors provide a more detailed and in-depth discussion on the structural optimization of the ultra-redundant cable-driven manipulator.
Round 2
Reviewer 3 Report
目前,作者的修改可以帮助潜在读者提高论文的学术性。